# Analysis of the Factors Affecting China's Manufacturing Servitization from the Perspective of the Ecological Environment

**Hui Li [1,*]** **, Lixia Chu [1] and Xiaoyi Qin [2]**

1   School of Business, Liaocheng University, Liaocheng 252000, China
2   Bathus College of Future Agricultural Science and Technology, Qingdao Agriculture University, Qingdao 266109, China
*   Correspondence: lihui@lcu.edu.cn; Tel.: +86-188-010-82529

**Abstract:** Manufacturing servitization (MS) can effectively alleviate the contradiction between economic growth and ecological carrying capacity and can promote energy conservation and emission reduction in the manufacturing industry (MI). However, China's MI is in the primary stage of servitization and lacks sufficient ability to provide services; thus, the environmental benefits of MS are not obvious. Therefore, in the context of current pressure to normalize environmental protection, how servitization can drive low-carbon development in MI while taking into account economic development has become an important topic at present. Thus, this study constructs an evaluation index system of factors based on a driver–pressure–state–impact–response (DPSIR) model, and uses the decision-making trial and evaluation laboratory (DEMATEL) and interpretive structural modeling (ISM) model to analyze the causal relationship and key elements among the influencing factors. The results show that from the perspective of the ecological environment, many factors affect MS, although to varying degrees. Among them, the proportion of R&D personnel, input intensity, and the proportion of clean energy are the main factors. Based on the transmission mechanism among these factors, we propose two paths to realizing the service-oriented, low-carbon development of China's MI.

**Keywords:** servitization; environmental benefits; influencing factors

## 1. Introduction

With the acceleration of the industrialization process, China is facing increasingly serious environmental problems such as environmental pollution, resource shortages, and ecological deterioration, which have seriously affected people's lives and have a negative impact on the construction of an ecological civilization. In addition, the growing global call for environmental protection has put enormous pressure on China's environmental and resource problems, and promoting the green development of industry has become a major practical problem that needs to be solved in China.

Manufacturing servitization (MS) refers to the transformation of the manufacturing-centered value chain into a service-centered value chain and generally includes two levels: input servitization and output servitization. In input (output) servitization, service elements (service products) occupy an increasingly important position in the total input (output) of manufacturing industry (MI). As an innovative development manufacturing mode, MS can reduce the dependence of enterprises on resources and energy, alleviate the contradiction between economic growth and ecological carrying capacity [1–5], and has become an important means to upgrade MI and enhance competitiveness. However, as China's MI is in the primary stage of servitization, most of its MI lacks sufficient ability to extend the value chain and provide integrated services, which indicates that the environmental benefits (EBs) of MS are not obvious. Therefore, it is critical to identify the key influencing

factors (IFs) of MS from the perspective of the ecological environment (EE) in order to promote green transformation and upgradethe MI.

So far, the relevant research on MS mainly focuses on consequence analysis, IFs, and realization paths. In research on consequence analysis, Kastalli and Van [6] posited that there are initial short-term gains from MS, but there are also barriers to profitability. Most scholars agree that MS can drive a high-quality development of China's economy [7–11]. In terms of IFs, scholars' opinions differ. Chen and Wang [12] believed that the characteristics of different top management teams have different effects on the service transformation of MI; Huang and Huo [13], based on international input–output data, concluded that independent innovation capabilities, a fair competition environment, service innovation capabilities, and human capital levels are key factors for improving MS. Li et al. [14] stressed that the service transformation of MI is the result of interaction among multiple factors. Zhou and Bao [15] posited that the opening of the digital service market can significantly elevate MS. Xiao et al. [16] argued that human capital wages, information technology investment, and good customer relationships are positively correlated with enterprise service innovation performance. Qi [17] indicated that the intensity of industrial competition, technical level, degree of servitization, and resource constraints have the most significant effect on MS from the perspective of organizational factors. Qi et al. [18] demonstrated that government policy, service economic level, and the openness of an industry's import service trade have an impact on MS. In addition, Tong and Zhang [19] emphasized that the opening policy of producer services can significantly promote the service of China's MI.

In terms of the MS development path, Zhang et al. [20] proposed that the MI could improve profits through the operation strategy of live broadcast service. Zhang et al. [21] compared the advantages and disadvantages of manufacturers remanufacturing waste products with manufacturers' authorized retailers remanufacturing waste products and concluded that the manufacturer remanufacturing mode is more beneficial to supply chain members in the absence of government funding policies. Jovanovic et al. [22] proposed three different development paths by analyzing a large multinational manufacturer. Su et al. [23] discussed four different implementation paths of service-embedded manufacturing from the perspective of the ecological service coefficient. Tian et al. [24] discussed the servitization path of four textile and garment manufacturing enterprises in China. Although these previous studies conducted a comprehensive analysis of MS, none linked the EBs. Under the current dual pressures of carbon neutrality and environmental protection, the EBs of services will also affect enterprises' decision making.

In addition, most scholars have demonstrated the EBs of MS from different industry perspectives, including the textile industry [25,26], garment enterprises [27], and the electronics and automobile industries [28]. Mont [29] indicated that the transformation of the industrial economy into a service economy can reduce the energy and resource input in production and yield better EBs. Glatt et al. [30] believed that a technical product-service system positively affects ecological sustainability and validated this result using agricultural machinery as an example. Wang et al. [4] stated that improving the service level of the MI can promote its green development. Zhu et al. [3] used panel data to demonstrate the positive energy-saving effect of MS.

Although the EBs of MS are considered in these papers, there are still many problems that deserve further attention. First, the existing studies analyze the IFs of MS from an economic perspective without considering the effect of the promotion of environmental protection on MS. Second, although the existing research has considered the influencing factors of manufacturing servitization, it lacks an in-depth analysis of the interaction and relationship between the IFs. To compensate for the above deficiencies, this study takes MI as the research object to analyze the IFs of MS, and establishes the multi-level hierarchical structure model of factors affecting MS using the decision making trial and evaluation laboratory (DEMATEL) and interpretive structural modeling (ISM) model. By calculating the centrality and the cause degree, the result factors and causal factors affecting MS are determined. In addition, through

the statistical analysis of data, this study deeply investigates the relationships among the factors that affect MS. The research results provide a reference for accelerating the MS process.

## 2. Analysis of the Factors Influencing MS

As mentioned earlier, the economic and environmental benefits of MS have been recognized by the majority of researchers. However, the degree of environmental improvement provided by MS varies according to the level of its development. Therefore, the driver–pressure–state–impact–response (DPSIR) model, which can simultaneously cover economic and environmental factors, was selected to analyze the IFs of MS.

### 2.1. DPSIR Model

The DPSIR model is a concept model of an evaluation index system (EIS) that was put forward and developed by the United Nations in the early 1990s and is widely used in environmental systems. The model simultaneously covers four major elements: economy, society, environment, and policy. It not only shows the impact of economic development on ecology but also reflects the feedback of the ecological environment to society.

The DPSIR model includes five parts: driving force (D), pressure (P), state (S), impact (I) and response (R). Each part also includes several lower-level indicators. Among them, "D" is the potential cause of environmental change, generally including social economy, industrial structure, and other indicators; "P" refers to the impact of human production activities on the EE; "S" refers to the state of the EE under the "P"; "I" refers to the changes in human health and socio-economic structure caused by the "S" of the environment; and "R" refers to the response taken by the government, organization, or individuals to mitigate environmental degradation.

### 2.2. Construction of the EIS Based on DPSIR Model

The MI is the key object of China's environmental control. Its development status and response to environmental control will affect the choice of national green development path. MS is not only an internal demand for the transformation, upgrading, and sustainable development of the MI but also the development path under the external pressure of creating an EE. Therefore, constructing the EIS of factors that affect MS from the five aspects of DPSIR can comprehensively reflect the two-way role of natural and socio-economic factors in the manufacturing ecosystem and seek an effective means for the high-quality development of the MI. Based on the meaning of each factor in the DPSIR model and the relevant literature, this study constructs an EIS that includes 12 indicators (Table 1). See Appendix A Table A1 for the specific meaning of each indicator.

**Table 1.** The EIS of IFs of MS.

| | Factors | Code | Supporting Literatures |
|---|---|---|---|
| Driving force (D) | Proportion of service revenue | F1 | Liu and Li [31], Liu [32] |
| | Growth rate of industrial added value | F2 | Wang and Zhang [33], Shi and Tong [34], Yang and Huang [35] |
| | Proportion of the high-tech MI | F3 | Yang and Huang [35] |
| Pressure (P) | Energy consumption intensity | F4 | Wang and Zhang [33], Shi and Tong [34] |
| | Pollutant emission intensity | F5 | Shi and Tong [34], Yang and Huang [35] |
| State (S) | Proportion of R&D personnel | F6 | Shi and Tong [34] |
| | Input intensity | F7 | Liu [32] |
| | Proportion of clean energy | F8 | Shi and Tong [34] |
| Impact (I) | Pollutant emission compliance rate | F9 | Wang and Zhang [33] |
| Response (R) | Proportion of R&D expenditure | F10 | Yang and Huang [35] |
| | Proportion of training funds | F11 | Yang and Huang [35] |
| | Proportion of environmental governance investment | F12 | Shi and Tong [34], Jiao et al. [36] |

### 2.2.1. Driving Force (D) Analysis

The driving force refers to the main economic factors leading to the instability of the ecosystem, including the proportion of service income, the growth rate of industrial added value, and the proportion of high-tech MI. Among them, the proportion of service income and the proportion of high-tech MI reflect the degree of optimization of the industrial structure of the MI and also indirectly reflect the inherent drive of the MI to improve the EE by saving energy. The growth rate of industrial added value reflects the macro-environment of the MI in economic development and also reflects the intermediate consumption of raw materials and fuel power by production activities.

### 2.2.2. Pressure (P) Analysis

The pressure index is the direct impact of the economic development of the MI on the ecosystem. It can be described from the perspectives of resource consumption and environmental pollution, including energy consumption intensity and pollutant emission intensity.

### 2.2.3. State (S) Analysis

The state is the actual situation of the ecosystem and economic development under the joint action of driving force and pressure. This index mainly includes the proportion of R&D personnel, input intensity, and the proportion of clean energy. The proportion of R&D personnel and input intensity reflect the resource distribution structure of the MI and indirectly reflect its impact on the environment and economic benefits. The proportion of clean energy reflects the degree of balance between energy consumption and the EE of the MI.

### 2.2.4. Impact (I) Analysis

The state of the EE is closely related to people's lives, and the constant changes in its state will have various impacts on production and life as well as on the EE itself. Among them, pollutant emission compliance rates and the ambient air quality all reflect the impact of the development of the MI on the EE. Considering the principle of indicator representativeness and avoiding repetition, this study selected the pollutant emission compliance rate as the impact indicator.

### 2.2.5. Response (R) Analysis

To realize the coordinated development of a manufacturing economy and the environment, human beings must adjust the development plan, and the process of adjustment is the process of response. The indicators that characterize the response mainly include the proportion of R&D expenditure, training funds, and environmental governance investment. The proportion of R&D and training expenditures reflect the strength of the MI in alleviating environmental pressure; the proportion of environmental treatment investment indirectly reflects the development of environmental pollution treatment.

## 3. Methods and Processing

### 3.1. DEMATEL–ISM Method

The decision making trial and evaluation laboratory (DEMATEL) is a systematic analysis method that uses graph theory and matrix tools [37]. It aims to effectively capture the causal relationship between factors by analyzing the logical relationship and direct influence matrix (IM) of each indicator. The interpretive structural modeling (ISM) model is often used in the structural analysis of complex systems [38]. It extracts the constituent elements of a problem, divides them into different levels, and uses the matrix operation to obtain the influence relationship between the elements to establish a structural model of the complex system.

Given certain similarities between DEMATEL and ISM methods, the DEMATEL–ISM integration method, which integrates the advantages of the two methods, has been proposed [39,40]. In this method, DEMATEL is initially used to sort the indicators according

to the causal degree and centrality of IFs to clarify the relative importance of these factors. The ISM is then used to divide the factors into different levels.

Considering that the factors influencing MS include factors such as economic development level, industrial structure, and energy consumption intensity, these factors might exhibit some direct and indirect influence relationships. Therefore, the DEMATEL–ISM integration method can be applied to analyze the IFs of the MS. The specific steps are as follows:

Step 1: Determine the IFs of MS according to the research problems and set them as $F_1, F_2, \cdots, F_n$.

Step 2: Construct a direct IM $X$. Using expert knowledge and experience to evaluate the direct influence relationship between different factors, the direct IM of order is obtained as follows:

$$X = \begin{bmatrix} 0 & x_{12} & \cdots & x_{1n} \\ x_{21} & 0 & \cdots & x_{2n} \\ \vdots & \vdots & \cdots & \vdots \\ x_{n1} & x_{n2} & \cdots & x_{nn} \end{bmatrix}$$

where $x_{ij}$ indicates the direct influence degree of factor $F_i$ on $F_j$.

Step 3: Normalize the direct IM as follows:

$$Z = X / \prod_{1 \leq i \leq n} \max \sum_{j=1}^{n} x_{ij} \tag{1}$$

Step 4: Calculate a comprehensive IM $T$ of the system:

$$T = \lim_{m \to \infty} (z + z^2 + \cdots z^m) = Z(I - Z)^{-1} \tag{2}$$

where $I$ is the identity matrix.

Step 5: Calculate the influence degree ($D$), affected degree ($Q$), centrality degree ($P$), and causal degree ($E$) of each factor:

$$D_i = \sum_{j=1}^{n} t_{ij} \tag{3}$$

$$Q_j = \sum_{i=1}^{n} t_{ij} \tag{4}$$

$$P_i = \left\{ D_i + Q_j \middle| i = j \right\} \tag{5}$$

$$E_i = \left\{ D_i - Q_j \middle| i = j \right\} \tag{6}$$

Step 6: Calculate the overall IM $H$:

$$H = I + T \tag{7}$$

Step 7: Select an appropriate threshold to calculate the reachability matrix $K$:

$$k_{ij} = \begin{cases} 1, t_{ij} \geq \lambda \\ 0, t_{ij} < \lambda \end{cases} \tag{8}$$

Step 8: Determine the reachable set $R_i$ and antecedent set $A_i$ of each influencing factor $F_i$:

$$R_i = \left\{ F_j \in F, k_{ij} \neq 0 \right\}, (i = 1, 2, \cdots, n) \tag{9}$$

$$A_i = \left\{ F_j \in F, k_{ji} \neq 0 \right\} \tag{10}$$

Step 9: Verify whether formula (11) is true, which indicates that the corresponding factor $F_i$ is the top-level factor, and delete the $i$-th row and $i$-th column in the matrix $K$:

$$R_i = R_i \cap A_i \tag{11}$$

Step 10: Repeat steps 8 and 9 until all factors are crossed out.

Step 11: Establish a hierarchical structure of factors according to the order of the crossed-out factors.

### 3.2. Data Collection

According to the indicators listed in Table 1, the original data were obtained through expert investigation (Appendix A Tables A2–A6). In this study, an expert team was formed by selecting five individuals from the fields of manufacturing and environmental protection, including one from academic experts (Shandong University of Technology) and four from enterprises, such as Yongxin Group (Beijing, China) and Nord China Transmission Equipment Co., Ltd. (Suzhou, China). (Appendix A Table A7). The average value of the experts' scoring was calculated as the initial direct influence matrix (Table 2).

**Table 2.** Initial direct IM.

| Code | F1 | F2 | F3 | F4 | F5 | F6 | F7 | F8 | F9 | F10 | F11 | F12 |
|------|-----|-----|-----|-----|-----|-----|-----|-----|-----|-----|-----|-----|
| F1 | 0 | 3.8 | 4 | 4.5 | 4.7 | 3.8 | 7.6 | 3.1 | 4.1 | 5.2 | 4.2 | 5.1 |
| F2 | 3.4 | 0 | 7.5 | 5.3 | 3.9 | 3.8 | 3.5 | 2.4 | 3.7 | 4.7 | 3.1 | 3.1 |
| F3 | 7.4 | 7.1 | 0 | 5.7 | 4.1 | 5.6 | 2.4 | 5.5 | 5.1 | 6.2 | 4.9 | 3.3 |
| F4 | 1.9 | 4.9 | 5.4 | 0 | 7 | 1.3 | 3.2 | 5.7 | 6.5 | 6 | 2.3 | 5.2 |
| F5 | 2.5 | 2.7 | 5.3 | 6.8 | 0 | 2 | 2.1 | 4.1 | 7.6 | 2.4 | 2.2 | 6.7 |
| F6 | 6.4 | 4.8 | 6 | 1.9 | 1 | 0 | 3 | 2 | 1.8 | 8.1 | 5.8 | 2.4 |
| F7 | 7.4 | 4.5 | 4.4 | 2.2 | 3.5 | 3 | 0 | 1.2 | 3.2 | 4.4 | 4 | 3.2 |
| F8 | 3.1 | 3.2 | 5.2 | 8.4 | 7.7 | 3.4 | 2.8 | 0 | 9.8 | 4.9 | 2.4 | 6 |
| F9 | 2.1 | 3.5 | 4.5 | 4.9 | 6.8 | 3 | 2.6 | 6.6 | 0 | 2.8 | 2.8 | 6.4 |
| F10 | 7 | 5.7 | 7 | 4.7 | 2.2 | 7.5 | 4.4 | 3.3 | 6 | 0 | 3.9 | 2.8 |
| F11 | 5.2 | 2.5 | 3.9 | 3.1 | 2 | 3.8 | 3.4 | 2.4 | 2.6 | 3.5 | 0 | 1.2 |
| F12 | 2.5 | 2.7 | 3.9 | 4.8 | 6.5 | 3.2 | 6.4 | 4.6 | 1.4 | 3 | 2.4 | 0 |

### 3.2.1. Calculation of Comprehensive IM

According to the data presented in Table 2, the comprehensive IM T (Table 3) was calculated using Formulas (1) and (2). The influence, affected, centrality, and causal degrees of each factor were calculated using Formulas (3)–(6) (Table 4). Figure 1 shows the causality picture of the IFs of MS.

**Table 3.** Comprehensive influence matrix.

| Code | F1 | F2 | F3 | F4 | F5 | F6 | F7 | F8 | F9 | F10 | F11 | F12 |
|------|------|------|------|------|------|------|------|------|------|------|------|------|
| F1 | 0.35 | 0.39 | 0.47 | 0.44 | 4.7 | 0.36 | 0.39 | 0.43 | 0.45 | 0.45 | 0.34 | 0.42 |
| F2 | 0.37 | 0.31 | 0.49 | 0.44 | 0.40 | 0.33 | 0.31 | 0.39 | 0.41 | 0.41 | 0.30 | 0.36 |
| F3 | 0.52 | 0.50 | 0.48 | 0.54 | 0.49 | 0.44 | 0.36 | 0.55 | 0.53 | 0.53 | 0.40 | 0.45 |
| F4 | 0.39 | 0.42 | 0.51 | 0.41 | 0.50 | 0.33 | 0.33 | 0.51 | 0.51 | 0.47 | 0.32 | 0.44 |
| F5 | 0.37 | 0.37 | 0.48 | 0.49 | 0.37 | 0.32 | 0.30 | 0.50 | 0.50 | 0.40 | 0.30 | 0.44 |
| F6 | 0.43 | 0.38 | 0.47 | 0.38 | 0.34 | 0.28 | 0.31 | 0.40 | 0.38 | 0.46 | 0.35 | 0.34 |
| F7 | 0.41 | 0.35 | 0.41 | 0.36 | 0.36 | 0.30 | 0.24 | 0.35 | 0.35 | 0.38 | 0.30 | 0.34 |
| F8 | 0.44 | 0.43 | 0.55 | 0.58 | 0.55 | 0.39 | 0.36 | 0.46 | 0.46 | 0.50 | 0.35 | 0.49 |
| F9 | 0.36 | 0.37 | 0.47 | 0.46 | 0.47 | 0.33 | 0.30 | 0.50 | 0.50 | 0.40 | 0.31 | 0.43 |
| F10 | 0.50 | 0.46 | 0.56 | 0.50 | 0.44 | 0.45 | 0.38 | 0.49 | 0.49 | 0.41 | 0.37 | 0.42 |
| F11 | 0.33 | 0.28 | 0.35 | 0.32 | 0.29 | 0.27 | 0.25 | 0.31 | 0.31 | 0.32 | 0.20 | 0.26 |
| F12 | 0.32 | 0.32 | 0.40 | 0.40 | 0.41 | 0.29 | 0.27 | 0.42 | 0.35 | 0.35 | 0.26 | 0.28 |

**Table 4.** The influence, affected, centrality, and causal degrees of each factor.

| Code | $D_i$ | $Q_i$ | $P_i$ | $E_i$ | Category |
|------|-------|-------|-------|-------|----------|
| F1 | 4.93 (5) | 4.78 (7) | 9.71 (7) | 0.149 (5) | Net cause |
| F2 | 4.53 (8) | 4.59 (9) | 9.11 (8) | −0.062 (7) | Net effect |
| F3 | 5.78 (1) | 5.64 (1) | 11.42 (1) | 0.146 (6) | Net cause |
| F4 | 5.13 (4) | 5.33 (2) | 10.46 (4) | −0.191 (8) | Net effect |
| F5 | 4.51 (9) | 4.09 (10) | 8.60 (10) | 0.427 (1) | Net cause |
| F6 | 4.83 (6) | 5.05 (6) | 9.89 (6) | −0.22 (9) | Net effect |
| F7 | 4.17 (10) | 3.80 (12) | 7.97 (11) | 0.375 (4) | Net cause |
| F8 | 5.69 (2) | 5.32 (3) | 11.01 (2) | 0.376 (3) | Net cause |
| F9 | 4.78 (7) | 5.31 (4) | 10.09 (5) | −0.535 (11) | Net effect |
| F10 | 5.50 (3) | 5.07 (5) | 10.57 (3) | 0.426 (2) | Net cause |
| F11 | 3.49 (12) | 3.80 (11) | 72.9 (12) | −0.583 (10) | Net effect |
| F12 | 4.08 (11) | 4.67 (8) | 8.74 (9) | −0.72 (12) | Net effect |

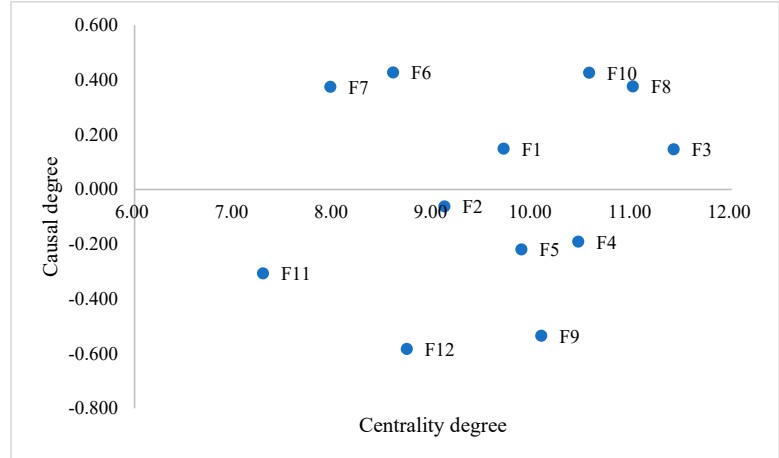

**Figure 1.** Causality diagram of the factors influencing manufacturing servitization.

3.2.2. Calculation of Reachability Matrix

Formula (7) was used to calculate the overall IM. To facilitate the analysis, a threshold ($\lambda = 0.395$) was selected after multiple experiments to eliminate relatively unimportant influences. The reachability matrix of the factors was calculated using the ISM method (Table 5), and the 12 factors were divided into different levels. The specific level division process is presented in Table 6. Figure 2 shows the multilevel hierarchical structure model obtained.

**Table 5.** Final reachability matrix.

| Code | F1 | F2 | F3 | F4 | F5 | F6 | F7 | F8 | F9 | F10 | F11 | F12 |
|------|----|----|----|----|----|----|----|----|----|-----|-----|-----|
| F1 | 1 | 0 | 1 | 1 | 1 | 0 | 0 | 1 | 1 | 1 | 0 | 1 |
| F2 | 0 | 1 | 1 | 1 | 1 | 0 | 0 | 0 | 1 | 1 | 0 | 0 |
| F3 | 1 | 1 | 1 | 1 | 1 | 1 | 0 | 0 | 1 | 1 | 1 | 1 |
| F4 | 0 | 1 | 1 | 1 | 1 | 0 | 0 | 1 | 1 | 1 | 0 | 1 |
| F5 | 0 | 0 | 1 | 1 | 1 | 0 | 0 | 1 | 1 | 1 | 0 | 1 |
| F6 | 1 | 0 | 1 | 0 | 0 | 1 | 0 | 1 | 0 | 1 | 0 | 0 |
| F7 | 1 | 0 | 1 | 0 | 0 | 0 | 1 | 1 | 0 | 0 | 0 | 0 |
| F8 | 1 | 1 | 1 | 1 | 1 | 0 | 0 | 0 | 1 | 1 | 0 | 1 |
| F9 | 0 | 0 | 1 | 1 | 1 | 0 | 0 | 1 | 1 | 1 | 0 | 1 |
| F10 | 1 | 1 | 1 | 1 | 1 | 1 | 0 | 1 | 1 | 1 | 0 | 1 |
| F11 | 0 | 0 | 0 | 0 | 0 | 0 | 0 | 0 | 0 | 0 | 1 | 0 |
| F12 | 0 | 0 | 1 | 1 | 1 | 0 | 0 | 1 | 0 | 0 | 0 | 1 |

**Table 6.** Level identification process for the 12 factors.

| Factors | Reachability Set | Antecedent Set | Intersection Set | Level |
|---|---|---|---|---|
| | | Iteration 1 | | |
| 1 | 1, 3, 4, 6, 8, 9, 10, 12 | 1, 3, 5, 7, 8, 10 | 3, 8, 10 | |
| F2 | 2, 3, 4, 6, 9, 10 | 2, 3, 4, 8, 10 | 2, 3, 4, 10 | |
| F3 | 1, 2, 3, 4, 5, 6, 8, 9, 10, 11, 12 | 1, 2, 3, 4, 5, 6, 7, 8, 9, 10, 12 | 1, 2, 3, 4, 5, 6, 8, 9, 10, 12 | |
| F4 | 2, 3, 4, 6, 8, 9, 10, 12 | 1, 2, 3, 4, 6, 8, 9, 10, 12 | 2, 3, 4, 6, 8, 9, 10, 12 | 1 |
| F5 | 3, 4, 6, 8, 9, 10, 12 | 1, 2, 3, 4, 6, 8, 9, 10, 12 | 3, 4, 6, 8, 9, 10, 12 | 1 |
| F6 | 1, 3, 5, 8, 10 | 3, 5, 10 | 3, 5, 10 | |
| F7 | 1, 3, 7 | 7 | 7 | |
| F8 | 1, 2, 3, 4, 6, 8, 9, 10, 12 | 1, 3, 4, 5, 6, 8, 9, 10, 12 | 1, 3, 4, 6, 8, 9, 10, 12 | |
| F9 | 3, 4, 6, 8, 9, 10, 12 | 1, 2, 3, 4, 6, 8, 9, 10 | 3, 4, 6, 8, 9, 10 | |
| F10 | 1, 2, 3, 4, 5, 6, 8, 9, 10, 12 | 1, 2, 3, 4, 5, 6, 8, 9, 10 | 2, 3, 4, 6, 8, 9, 10 | |
| F11 | 11 | 3, 11 | 1, 2, 3, 5, 8, 9, 10 | 1 |
| F12 | 3, 4, 6, 8, 12 | 1, 3, 4, 6, 8, 9, 10, 12 | 3, 4, 6, 8, 12 | 1 |
| | | Iteration 2 | | |
| F1 | 1, 3, 8, 9, 10 | 1, 3, 5, 7, 8, 10 | 1, 3, 8, 10 | |
| F2 | 2, 3, 9, 10 | 2, 3, 8, 10 | 2, 3, 10 | 2 |
| F3 | 1, 2, 3, 5, 8, 9, 10 | 1, 3, 5, 8, 9, 10 | 1, 3, 8, 9, 10 | |
| F6 | 1, 3, 5, 8, 10 | 3, 5, 10 | 3, 5, 10 | |
| F7 | 1, 3, 7 | 7 | 7 | |
| F8 | 1, 2, 3, 8, 9, 10 | 1, 3, 8, 9, 10 | 1, 3, 8, 9, 10 | |
| F9 | 3, 8, 9, 10 | , 2, 3, 8, 9, 10 | 3, 8, 9, 10 | |
| F10 | 1, 2, 3, 5, 8, 9, 10 | 1, 2, 3, 5, 8, 9, 10 | 1, 2, 3, 5, 8, 9, 10 | 2 |
| | | Iteration 3 | | |
| F1 | 1, 8 | 1, 5, 7, 8 | 1, 8 | 3 |
| F2 | 2 | 2, 8 | 2 | 3 |
| F6 | 1, 5, 8 | 5 | 5 | |
| F7 | 1, 7 | 7 | 7 | 3 |
| F8 | 1, 2, 8 | 1, 5, 8 | 1, 8 | |
| | | Iteration 4 | | |
| F6 | 5, 8 | 5 | 5 | |
| F7 | 7 | 7 | 7 | 4 |
| F8 | 8 | 5, 8 | 8 | 4 |
| | | Iteration 5 | | |
| F6 | 5 | 5 | 5 | 5 |

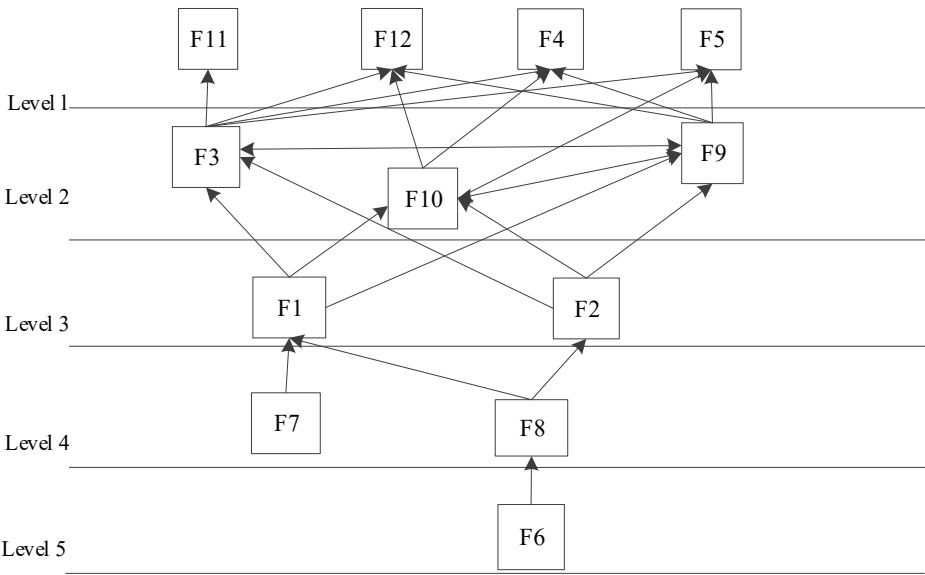

**Figure 2.** Hierarchical structure of influencing factors.

**4. Results and Discussion**

*4.1. Analysis of Results*

4.1.1. Impact Level Analysis

Based on the hierarchical structure model of the IFs of MS, the 12 factors can be divided into five levels (Figure 2). Level 1 captures the most direct and basic factors affecting MS, mainly including F4 (energy consumption intensity), F5 (pollutant emission intensity), F11 (proportion of training expenditure), and F12 (proportion of environmental governance investment). They are the internal factors affecting MS and play a role through the influence of lower-level factors. Therefore, in the process of promoting MS, we should not only achieve economic development but also reduce the impact on the environment, strengthen enterprise talent training, and accelerate the innovative green development of MI. Levels 2 and 3 are indirect factors affecting MS, mainly including F1 (proportion of service revenue), F2 (growth rate of industrial added value), F3 (proportion of high-tech MI), F9 (pollutant emission compliance rate), and F10 (proportion of R&D expenditure). They are influenced by the lower factors and act on the upper factors. Levels 4 and 5 are the most fundamental and strongest factors affecting MS. The levels harbor three factors: F7 (input intensity), F8 (proportion of clean energy), and F6 (proportion of R&D personnel). To promote MS, we should start from the root, increasing R&D investment, promoting the service trend of manufacturing investment, and improving the proportion of clean energy use, so as to promote the status of China's MI in the global industrial value chain.

4.1.2. Impact Path Analysis

According to the ISM model theory and hierarchy division, this study divided the realization path of the MS into two paths. Path 1 is $F_6 \rightarrow F_8 \rightarrow (F_1, F_2) \rightarrow (F_3, F_9, F_{10}) \rightarrow (F_4, F_5, F_{12})$, and path 2 is $F_7 \rightarrow F_1 \rightarrow (F_3, F_9, F_{10}) \rightarrow (F_4, F_5, F_{12})$. Path 1 shows that talent is the key to the generation of MS. The management department should start by increasing the number of R&D personnel, accelerating the development and utilization of clean energy, promoting the growth of servitization income and the industrial added value of the MI, and optimizing the industrial structure to reduce environmental pollution and improve air quality. This path highlights the importance of technology R&D and can be defined as a technology-driven development path. Path 2 is to increase the service income of MI by increasing the intensity of investment in manufacturing services, so as to promote the optimization and upgrading of MI structures and reduce environmental pollution. This path emphasizes the importance of the increasing intermediate investment of MI service trends and can be defined as a business-driven development path. This indicates that MS can be realized not only through the technology-driven development path but also through the business-driven development path.

*4.2. Policy Discussion*

Based on the above analysis results, this paper proposes the following suggestions for the current development of China's MI:

(1) Address the technical aspects of training and introduction. The Chinese government has always attached great importance to the cultivation and introduction of talent. The 20th National Congress of the Communist Party of China put forward a strategy of strengthening the country with talent, fully affirming the importance of talent for the development of the country and enterprises. Talent is not only the foundation of enterprise development but also an important driving force for MS. In the process of promoting MS, increasing the introduction and training of talented individuals can comprehensively improve the innovation ability of the MI, promote knowledge sharing and transfer, produce significant benefits of technology diffusion and knowledge spillover, and promote the creation of EBs of MS;

(2) Improve manufacturing input intensity. This is conducive to giving full play to the EBs of MS. At present, China's MI is facing new problems and challenges, such as increased uncertainty in the international market, rising costs of domestic production

factors, and tightening constraints on resources and the environment. Increasing manufacturing input intensity can not only encourage the MI to climb to both ends of the value chain and increase its added value but also reduce energy dependence and environmental pollution. Therefore, improving manufacturing input intensity is an inevitable requirement for the development of the MI and an important way to build an ecological civilization;

(3) Increase the proportion of clean energy consumption and optimize the energy structure. China's MI is an energy-intensive industry, and fossil energy is its main energy source. With the approaching national carbon peak and carbon-neutral target period, fossil energy consumption in the MI has become the main object of national concern. Therefore, in the process of MS, adhering to a clean and low-carbon orientation and accelerating the green and low-carbon transformation of energy is an important direction to fully capture the advantages of the EBs of MS as soon as possible;

(4) Leverage the dual-driven "technology and business" model to accelerate MS. According to the results of hierarchical structure analysis, MS can be realized in two ways: technology-driven and business-driven. In promoting the servitization process of China's MI, technological innovation can promote the MI and service industry to innovate production modes, business modes, and organization modes and produce environmental benefits. Similarly, more service factors in the MI can not only provide power for technological innovation but also play a positive role in energy conservation and emission reduction. Therefore, to speed up the realization of MS, technology-driven and business-driven modes need to operate together.

## 5. Conclusions

MS has become an important development trend in global industry and it is also the only way to transform China's MI. To accelerate capturing the EBs of MS and promote the high-quality development of the MI, this study analyzed the relationship between factors influencing MS and used the DEMATEL–ISM method to obtain the influences, affected centers, and cause degrees. On this basis, a hierarchical structure that defines the relationships among the factors was constructed. The following conclusions were drawn from the findings:

(1) In terms of the degree of impact, the factors affecting the EBs of MS can be divided into five different levels. The most important factor (Level 5) is the proportion of R&D personnel. Level 4 includes the proportion of clean energy. Level 3 is the proportion of service revenue and growth rate of industrial added value. Level 2 harbors the proportion of high-tech MI, pollutant emission compliance rate, and proportion of R&D expenditure. Level 1 includes energy consumption intensity, pollutant emission intensity, the proportion of training expenditure, and the proportion of environmental governance investment.

(2) According to the transmission mechanism of the factors, we propose two main methods to realize the EBs of MS from the external and internal driving mechanisms, aiming to jointly promote the realization of the EBs of MS through internal and external linkages.

The findings of this study are the results of a static analysis of the factors affecting MS; thus, the conclusion is only suitable for the current stage of the development of China's MS. With continuous improvement in the degree of development of MS, the influence relationship between various factors will also change. Therefore, a dynamic analysis of the factors influencing MS should be undertaken in the future.

**Author Contributions:** Conceptualization, H.L. and L.C.; methodology, H.L., L.C. and X.Q.; investigation, H.L. and X.Q.; data curation, X.Q.; writing—original draft preparation, H.L. and X.Q.; supervision, H.L. and L.C.; project administration, H.L. All authors have read and agreed to the published version of the manuscript.

**Funding:** This work was supported by the National Social Science Foundation of China (Grant No. 20BJY074), the Shandong Social Science Planning Foundation (Grant No. 21CJJJ14), and the Doctoral Scientific Research Foundation (Grant No. 321052027).

**Institutional Review Board Statement:** Not applicable.

**Informed Consent Statement:** Not applicable.

**Data Availability Statement:** Data available within the article.

**Acknowledgments:** Thanks to the comments and suggestions of the two reviewers and the editors, the quality of this article has been further improved.

**Conflicts of Interest:** The authors declare no conflict of interest.

## Appendix A

**Table A1.** Specific description of evaluation index.

| | Factor | Code | Description |
|---|---|---|---|
| Driving force (D) | Proportion of service revenue | F1 | Proportion of revenue from services provided by MI in total revenue |
| | Growth rate of industrial added value | F2 | Annual growth rate of industrial added value |
| | Proportion of the high-tech MI | F3 | Proportion of the output value of high-tech industry in the total output value of MI |
| Pressure (P) | Energy consumption intensity | F4 | Energy consumption per unit of industrial added value |
| | Pollutant emission intensity | F5 | Pollutant emissions per unit of industrial added value |
| State (S) | Proportion of R&D personnel | F6 | Ratio of the number of R&D personnel to the total number of manufacturing personnel |
| | Input intensity | F7 | Proportion of manufacturing service investment in total manufacturing investment |
| | Proportion of clean energy | F8 | Proportion of clean energy in total energy consumption |
| Impact (I) | Pollutant emission compliance rate | F9 | Ratio of pollutant emission to total emission |
| Response (R) | Proportion of R&D expenditure | F10 | Proportion of R&D expenditure in total industrial output value |
| | Proportion of training expenditure | F11 | Proportion of training funds invested by MI in the total investment |
| | Proportion of environmental governance investment | F12 | Proportion of environmental governance investment in total output value of MI |

**Table A2.** Raw data of expert 1.

| Code | F1 | F2 | F3 | F4 | F5 | F6 | F7 | F8 | F9 | F10 | F11 | F12 |
|---|---|---|---|---|---|---|---|---|---|---|---|---|
| F1 | 0 | 4 | 5 | 5 | 9 | 6 | 10 | 4 | 6 | 9 | 8 | 4 |
| F2 | 1 | 0 | 8 | 8 | 6 | 8 | 2 | 0 | 2 | 8 | 6 | 4 |
| F3 | 8 | 7 | 0 | 8 | 6 | 6 | 7 | 8 | 5 | 6 | 6 | 4 |
| F4 | 0 | 0 | 4 | 0 | 10 | 0 | 2 | 9 | 10 | 3 | 0 | 5 |
| F5 | 0 | 0 | 3 | 10 | 0 | 4 | 2 | 9 | 0 | 4 | 4 | 8 |
| F6 | 8 | 5 | 6 | 3 | 0 | 0 | 6 | 5 | 4 | 10 | 8 | 4 |
| F7 | 8 | 3 | 3 | 5 | 5 | 8 | 0 | 0 | 4 | 6 | 6 | 3 |
| F8 | 4 | 0 | 5 | 10 | 10 | 8 | 2 | 0 | 9 | 8 | 6 | 8 |
| F9 | 0 | 0 | 2 | 9 | 0 | 5 | 0 | 9 | 0 | 4 | 2 | 9 |
| F10 | 8 | 4 | 8 | 6 | 6 | 10 | 8 | 4 | 6 | 0 | 2 | 6 |
| F11 | 8 | 2 | 4 | 6 | 4 | 6 | 8 | 2 | 2 | 6 | 0 | 0 |
| F12 | 2 | 1 | 0 | 8 | 8 | 6 | 2 | 6 | 9 | 4 | 1 | 0 |

**Table A3.** Raw data of expert 2.

| Code | F1 | F2 | F3 | F4 | F5 | F6 | F7 | F8 | F9 | F10 | F11 | F12 |
|------|----|----|----|----|----|----|----|----|----|-----|-----|-----|
| F1 | 0 | 7 | 3 | 8 | 5 | 5 | 5 | 2 | 2 | 3 | 3 | 8 |
| F2 | 6 | 0 | 7 | 3 | 3 | 2 | 5 | 3 | 3 | 2 | 3 | 3 |
| F3 | 6 | 5 | 0 | 2 | 3 | 3 | 2 | 3 | 3 | 6 | 4 | 4 |
| F4 | 2 | 4 | 5 | 0 | 3 | 2 | 2 | 2 | 2 | 3 | 4 | 4 |
| F5 | 4 | 3 | 5 | 3 | 0 | 3 | 2 | 2 | 6 | 4 | 3 | 3 |
| F6 | 8 | 3 | 5 | 2 | 2 | 0 | 3 | 3 | 2 | 9 | 3 | 4 |
| F7 | 6 | 6 | 6 | 3 | 3 | 3 | 0 | 2 | 2 | 4 | 2 | 3 |
| F8 | 4 | 3 | 4 | 6 | 9 | 3 | 3 | 0 | 3 | 5 | 2 | 6 |
| F9 | 3 | 2 | 4 | 3 | 3 | 3 | 6 | 3 | 0 | 4 | 2 | 2 |
| F10 | 8 | 4 | 6 | 4 | 2 | 6 | 2 | 4 | 3 | 0 | 3 | 4 |
| F11 | 5 | 2 | 6 | 3 | 2 | 3 | 2 | 3 | 3 | 4 | 0 | 4 |
| F12 | 3 | 2 | 6 | 5 | 4 | 2 | 6 | 2 | 4 | 4 | 2 | 0 |

**Table A4.** Raw data of expert 3.

| Code | F1 | F2 | F3 | F4 | F5 | F6 | F7 | F8 | F9 | F10 | F11 | F12 |
|------|----|------|----|----|----|----|----|----|----|-----|-----|-----|
| F1 | 0 | 2.5 | 4 | 2 | 2 | 3 | 8 | 2 | 2 | 5 | 2 | 5 |
| F2 | 5 | 0 | 8 | 5 | 3 | 2 | 5 | 3 | 3 | 5 | 3 | 3 |
| F3 | 8 | 8 | 0 | 3 | 3 | 3 | 2 | 8 | 3 | 5 | 5 | 5 |
| F4 | 1 | 5 | 5 | 0 | 4 | 1 | 2 | 8 | 3 | 5 | 4 | 4 |
| F5 | 2 | 3 | 5 | 4 | 0 | 2 | 2 | 9 | 8 | 3 | 3 | 8 |
| F6 | 5 | 2 | 3 | 1 | 2 | 0 | 3 | 3 | 2 | 8 | 5 | 3 |
| F7 | 8 | 5 | 6 | 2 | 2 | 3 | 0 | 3 | 2 | 7 | 2 | 2 |
| F8 | 2 | 3 | 4 | 8 | 8 | 3 | 8 | 0 | 3 | 5 | 4 | 3 |
| F9 | 2 | 3 | 3 | 3 | 2 | 2 | 2 | 3 | 0 | 3 | 2 | 5 |
| F10 | 8 | 5 | 5 | 5 | 2 | 6 | 5 | 5 | 3 | 0 | 5 | 4 |
| F11 | 4 | 3 | 5 | 4 | 3 | 5 | 2 | 4 | 2 | 5 | 0 | 2 |
| F12 | 2 | 3 | 5 | 4 | 5 | 3 | 2 | 6 | 5 | 4 | 2 | 0 |

**Table A5.** Raw data of expert 4.

| Code | F1 | F2 | F3 | F4 | F5 | F6 | F7 | F8 | F9 | F10 | F11 | F12 |
|------|----|----|----|----|----|----|----|----|----|-----|-----|-----|
| F1 | 0 | 3 | 3 | 5 | 5 | 2 | 7 | 5 | 6 | 4 | 3 | 6 |
| F2 | 1 | 0 | 7 | 3 | 5 | 2 | 3 | 1 | 3 | 1 | 1 | 3 |
| F3 | 7 | 8 | 0 | 8 | 1 | 8 | 1 | 5 | 6 | 6 | 7 | 1 |
| F4 | 4 | 8 | 7 | 0 | 8 | 1 | 5 | 7 | 9 | 9 | 1 | 8 |
| F5 | 4 | 5 | 6 | 7 | 0 | 1 | 2 | 8 | 7 | 1 | 1 | 7 |
| F6 | 6 | 9 | 8 | 1 | 1 | 0 | 3 | 1 | 1 | 6 | 8 | 1 |
| F7 | 7 | 6 | 7 | 1 | 5 | 1 | 0 | 4 | 7 | 5 | 5 | 8 |
| F8 | 3 | 8 | 8 | 9 | 9 | 1 | 1 | 0 | 9 | 4 | 0 | 8 |
| F9 | 3 | 5 | 5 | 2 | 8 | 6 | 5 | 8 | 0 | 3 | 8 | 6 |
| F10 | 6 | 8 | 6 | 6 | 1 | 8 | 7 | 1 | 0 | 0 | 7 | 0 |
| F11 | 5 | 3 | 2 | 0 | 1 | 0 | 0 | 5 | 0 | 0 | 0 | 0 |
| F12 | 3 | 5 | 6 | 2 | 8 | 5 | 4 | 8 | 6 | 3 | 7 | 0 |

**Table A6.** Raw data of expert 5.

| Code | F1 | F2 | F3 | F4 | F5 | F6 | F7 | F8 | F9 | F10 | F11 | F12 |
|------|----|----|----|----|----|----|----|----|----|-----|-----|-----|
| F1 | 0 | 2.5 | 5 | 2.5 | 2.5 | 3 | 8 | 2.5 | 2.5 | 5 | 5 | 2.5 |
| F2 | 4 | 0 | 7.5 | 7.5 | 2.5 | 5 | 2.5 | 5 | 7.5 | 7.5 | 2.5 | 2.5 |
| F3 | 8 | 7.5 | 0 | 7.5 | 7.5 | 8 | 0 | 7.5 | 7.5 | 8 | 2.5 | 2.5 |
| F4 | 2.5 | 7.5 | 6 | 0 | 10 | 2.5 | 5 | 8 | 7.5 | 10 | 2.5 | 5 |
| F5 | 2.5 | 2.5 | 7.5 | 10 | 0 | 0 | 2.5 | 8 | 0 | 0 | 0 | 7.5 |
| F6 | 5 | 5 | 8 | 2.5 | 0 | 0 | 0 | 5 | 0 | 7.5 | 5 | 0 |
| F7 | 8 | 2.5 | 0 | 0 | 2.5 | 0 | 0 | 0 | 0 | 0 | 5 | 0 |
| F8 | 2.5 | 2 | 5 | 9 | 2.5 | 2 | 0 | 0 | 10 | 2.5 | 0 | 5 |
| F9 | 2.5 | 7.5 | 7.5 | 7.5 | 0 | 0 | 0 | 10 | 0 | 0 | 0 | 10 |
| F10 | 5 | 7.5 | 10 | 2.5 | 0 | 7.5 | 0 | 6 | 0 | 0 | 2.5 | 0 |
| F11 | 4 | 2.5 | 2.5 | 2.5 | 0 | 5 | 5 | 0 | 0 | 2.5 | 0 | 0 |
| F12 | 2.5 | 2.5 | 2.5 | 5 | 7.5 | 0 | 0 | 10 | 10 | 0 | 0 | 0 |

**Table A7.** Experts' information.

| Expert | Designation | Field | Experience Experts' Information (yrs) |
|--------|-------------|-------|----------------------------------------|
| Expert 1 (Qing Yang) | Production manager | Production management | 13 |
| Expert 2 (Pengtao Liu) | Director of production department | Organization and arrangement of production | 20 |
| Expert 3 (Guoan Liu) | Senior executive | Business process reengineering | 11 |
| Expert 4 (Shujiao Yi) | Information manager | Lean management | 18 |
| Expert 5 (Meixiang Wu) | Professor | Sustainable development | 25 |

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
