# Peer review of "Analysis of the Factors Affecting China’s Manufacturing Servitization from the Perspective of the Ecological Environment"

_sustainability, doi:10.3390/su15042934_

Round 1
Reviewer 1 Report
Thank you for the opportunity to review your manuscript. I appreciate your efforts in the work put into the analysis and presentation of the information concerning the analysis on the factors affecting China’s manufacturing servitization from the perspective of ecological environment. The paper provides an interesting analysis of the topic.
To enhance the quality of the study, it would be wise to pay attention to several issues:
- The introduction should clearly state the research question(s) or the research objective(s) that the authors present for this paper.
- The Literature review section that could bring support to the information provided in the paper is very limited. The authors should expand this section. Some references that could be included here could be:
o Cuc, S., Girneata, A., Iordanescu, M., & Irinel, M. (2015). Environmental and socioeconomic sustainability through textile recycling. Industria Textila, 66(3), 156-163.
o Girneata, A., & Mascu, M. (2014). Development discrepancies between Western and Eastern EU countries: a statistical analysis of textile and apparel clusters. In Proceedings of the 8th International Management Conference “Management challenges for the sustainable development”, Bucharest, Romania (pp. 434-442).
- Authors must clearly state who are the experts who participated in the study. Also, a description of the investigated population should be provided in the Methods and processing section.
- I recommend a different section to be included in the paper: “Findings”, where authors can present and detail the findings and importance of the study results.
- The “Conclusions” part must be developed more in order to summarize the main findings of the research, how the research objectives are met through this study and to whom are the results addressed.
- Authors should revise the grammar and punctuation of the paper. For example: at line 162, authors state: “First, the 11DEMATEL method is used to sort the indicators”.
- Authors also need to explain about study limitations in the revised draft and to include information regarding future research directions.
- The reference list is not formatted according to the MDPI guidelines and should be enhanced to include more recent or relevant studies.
Reviewer 2 Report
The current paper sheds light on the role of manufacturing servitization (MS) in the context of China manufacturing industry (MI) and its environmental and economic benefits. In particular, the study develops the evaluation index system (EIS) of factors following the conceptual model of DPISR (driving force, pressure, state, impact and response) and investigate the relationship between the influencing factors of MS and their influence on MS by employing the decision making trial and evaluation laboratory (DEMATEL) and interpretive structural modelling (ISM). The results highlight that the R&D personnel, the proportion of service revenue and the proportion of clean energy exert an influence of the China’s MS.
Furthermore, I would have several recommendations that can be directions for improving the manuscript:
- The introduction section should provide contributions of the study in comparison to previous studies. More specifically, the authors could clarify how the current empirical evidences contribute to advance prior studies. If addressed, this aspect may add clarity and value to the study.
- Regarding the data collection procedure, the authors could give more information about data sources and the object of the study. In particular, I suggest to clarify what is the object of the investigation by the experts in order to collect data. Moreover, the nature of the data collected is not specified. Indeed, it is unclear if the data are country-level. This issue needs to be addressed.
- Due to the focus on the context of China’s MS, I suggest to frame the MS phenomenon relying on Chinese MI background. For instance, it could be highlighted the stage of MS or the degree of environmental responsibility of Chinese MI and how the MS can provide environmental and economic benefits for the China MI context.
- The Appendix A does not report the Code “F8”.
- The limitations of the study are lacking.
Reviewer 3 Report
This paper attempts to investigate an interesting question about the how servitization can drive low-carbon development in MI while taking into account economic development. To obtain some novel findings, this paper constructs an evaluation index system of factors on basis of DPSIR (driving force, pressure, state, impact, and response) model, and uses the decision-making trial and evaluation laboratory and interpretive structural modeling methods to analyze the causal relationship and key elements among the influencing factors. Overall, this is an interesting research subject. However, there are some concerns as follows which should be applied:
1. In the Abstract section, the conclusion should be mentioned more briefly and clearly.
2. Introduction section should be strengthened strongly. The background of this paper is too short, which should be well-organized. In addition, the contributions of this work should be summarized more clearly in this section.
3. Literature review in the Introduction section is too weak, which should be improved. Literature review should not a simple stack of papers, but a comprehensive analysis. In addition, the authors should update the latest literatures in the literature review sections, such as doi: 10.1111/itor.13186 and doi: 10.1016/j.cie.2020.106951.
4. In the Analysis on the IFs of MS section, the main references of some concepts should be mentioned, such as DPSIR model.
5. In the Methods and processing section, there are a number of methods in this paper, it is a litter hard to evaluate which were developed by the authors and which parts were taken from published works.
6. Results and discussion section should be enhanced. The author should provide some practical examples or literature comparisons to support your results and findings in the Policy suggestions section, which may highlight your contributions.
7. The logic of this paper is not very clear, and the language should be improved strongly. Please check the formula symbols and typos in the paper to ensure that they are correct.
Round 2
Reviewer 2 Report
The current version of the paper provides several improvements in comparison to its previous one. The Introduction section has been revised in line with my suggestion. Indeed, the current version of the section highlights the contributions of the study in terms of advancement with respect to the extant literature in a clear way. With reference to the data collection paragraph, the authors have better specified the data sources and identified the object of the analysis. Finally, the limitations of the study are now provided.
Reviewer 3 Report
The paper has been revised according to the suggestions, and it is recommended to accept.